# Development of an Antigen Detection Kit Capable of Discriminating the Omicron Mutants of SARS-CoV-2

**DOI:** 10.3390/vaccines11020303

**Published:** 2023-01-30

**Authors:** Jiaji Li, Jinrong Shi, Zhijun Zhou, Bo Yang, Jiamin Cao, Zhongsen Cao, Qiang Zeng, Zheng Hu, Xiaoming Yang

**Affiliations:** 1School of Bioengineering and Food Science, Hubei University of Technology, Wuhan 430068, China; 2Wuhan Institute of Biological Products Co., Ltd., Wuhan 430207, China; 3VIDA-BIO Co., Ltd., Wuhan 430068, China; 4China National Biotec Group Company Limited, Beijing 100029, China; 5National Engineering Technology Research Center for Combined Vaccines, Wuhan 430207, China

**Keywords:** novel coronavirus, monoclonal antibody, colloidal gold immunochromatography, antigen detection kit

## Abstract

Introduction: Severe acute respiratory syndrome coronavirus 2 (SARS-CoV-2) has spread around the world, caused millions of deaths and a severe illness which poses a serious threat to human health. Objective: To develop an antigen detection kit that can identify Omicron novel coronavirus mutants. Methods: BALB/c mice were immunized with the nucleocapsid protein of SARS-CoV-2 Omicron mutant treated with β-propiolactone. After fusion of myeloma cells with immune cells, Elisa was used to screen the cell lines capable of producing monoclonal antibodies. The detection kit was prepared by colloidal gold immunochromatography. Finally, the sensitivity, specificity and anti-interference of the kit were evaluated by simulating positive samples. Results: The sensitivity of the SARS-CoV-2 antigen detection kit can reach 62.5 TCID_50_/mL, and it has good inclusiveness for different SARS-CoV-2 strains. The kit had no cross-reaction with common respiratory pathogens, and its sensitivity was still not affected under the action of different concentrations of interferences, indicating that it had good specificity and stability. Conclusion: In this study, monoclonal antibodies with high specificity to the N protein of the Omicron mutant strain were obtained by monoclonal antibody screening technology. Colloidal gold immunochromatography technology was used to prepare an antigen detection kit with high sensitivity to detect and identify the mutant Omicron strain.

## 1. Introduction

The novel coronavirus (SARS-CoV-2) has mutated continuously since its outbreak in 2019, and variants such as β, δ and Omicron have emerged [1,2]. Spike protein (S protein) and Nucleocapsid (N protein) are the two important structural proteins for the diagnosis and screening of COVID-19 [3]. A large number of studies and cases have shown that the mutation of S protein RBD of SARS-CoV-2 can significantly enhance the infectivity of SARS-CoV-2 to host cells, which may lead to enhanced transmission and virulence of SARS-CoV-2 [4]. N protein is abundant in coronavirus with high immunogenicity, which is also involved in viral genome replication and cell signaling pathway regulation. At the same time, the existing studies suggest that the N protein does not change much in multiple mutations [5,6,7]. Clinical results show that the N-antigen can be detected one day before the onset of clinical symptoms, so the N-protein is considered to be one of the best targets for detecting the novel coronavirus [8].

Currently, reverse transcriptase polymerase chain reaction (RT-PCR) detection of COVID-19 nucleic acid is the gold standard for clinical diagnosis of suspected cases [9], but this method needs at least a biosafety level 2 (BSL-2) requirement, professional equipment, and trained laboratory personnel; in addition, the operating process is complicated. Moreover, the test result can be easily affected by specimen quality, and the experiment resources in remote areas, communities or scenes are difficult to carry out, so using colloidal gold immune chromatography test paper to detect virus infection is another key method. The World Health Organization has recommended antigenic rapid diagnostic tests as an adjunct to the early diagnosis of SARS-CoV-2 infection. This method possesses the advantages of simple operation, rapid and low cost, and intuitive determination of the results.

In this study, N protein was used as the target antigen to immunize mice treated with β-propiolactone, and monoclonal antibodies with high sensitivity and good stability were prepared by hybridoma cell screening technology [10]. A rapid and accurate SARS-CoV-2 antigen detection kit was established by colloidal gold immunochromatography [11]. According to the test results of the kit, whether the patient is infected with the novel coronavirus can be determined as well as whether the infected strain is the Omicron mutant strain.

## 2. Materials and Methods

### 2.1. Preparation of Monoclonal Antibodies against SARS-CoV-2 N Protein

Mouse immunization: BALB/c mice were immunized with the recombinant N protein (Purchased from China Wuhan AtaGenix Biotechnology Co., Ltd. Item No: 618-092320-B01) solution mixed with β-propiolactone (volume ratio: 5000:1). When the antibody titer in mice reached the predetermined value (the antibody titer was measured at a dilution of 1:64,000), the N protein antigen was injected directly into the abdominal cavity of mice to enhance the immune effect [12]. After 3 days of immunization, spleen cells of immunized mice were fused with myeloma cells (SP2/0). Indirect enzyme-linked immunosorbent assay (Elisa) was used to screen out stable hybridoma cell lines with large amount of antibody secretion [13]. Monoclonal cell lines were cultured in the abdominal cavity of mice; serum was collected and purified to obtain monoclonal antibodies. After re-coating with N protein and inactivating virus, indirect Elisa was used again to screen monoclonal antibodies [14,15,16].

### 2.2. Antibody Pairing of Colloidal Gold Immunochromatography Strip

Colloidal gold immunochromatography test strips were prepared according to the method of references [17,18,19]. The antibodies were successfully paired to assemble the test strips, and the original strain, different types of SARS-CoV-2 mutants such as β, δ and Omicron mutants were verified.

### 2.3. Evaluation of Sensitivity of Colloidal Gold Immunochromatography Strip

Evaluation of the sensitivity of the colloidal gold strip was performed by detection of the inactivated SARS-CoV-2 culture. The culture of SARS-CoV-2 inactivated by β-propiolactone was diluted 10-fold with lysate and labeled as 10, 10^2^, 10^3^, 10^4^, 10^5^, 10^6^, 10^7^, and 10^8^. Then, the dipstick cards were sampled for each dilution to preliminarily confirm the detection limit. According to the color development of the test card, the 2-fold gradient series dilution was continued near the detection limit, and the color development degree of the C line and T line was observed. Finally, the lowest dilution concentration that could observe positive results was used as the lowest detection limit of the test card sensitivity [20].

The concentration of the inactivated SARS-CoV-2 culture was detected by the nucleic acid detection reagents. Briefly, the inactivated SARS-CoV-2 culture was serially diluted 10-fold with normal saline at a dilution of 1:10~1:10^10^. RNA was extracted by RNA extraction reagent and amplified by nucleic acid detection kit. The highest dilution that could observe positive results was used as the sensitivity of nucleic acid detection reagent, and the results were compared with the results of test card.

Due to the strong harm of virulent viruses, laboratories do not have the conditions for direct detection of viruses. Therefore, this study simulated the clinical detection experiment of infected patients by collecting nasal swab samples of normal people and adding diluted inactivated virus solution [21]. In this study, it was difficult to collect clinical patient samples when verifying mutant strains, so this method was used to simulate clinical human samples to complete the experiment. Colloidal gold test paper detection and nucleic acid detection experiments were completed by the scientific research and Development Department of China Wuhan Blood Products Co., Ltd., Sinopharm Group [22].

### 2.4. Preparation of a Test Paper for Identification of Omicron Mutant

Multiple sets of highly sensitive antibody pairs were selected, including those that could detect original strain, β, δ and Omicron mutant strains, and those that could not detect Omicron mutant strains. The two groups of different antibodies were prepared on a strip of test paper to produce a single card with two detection lines. The antibody that could not detect the Omicron strain on the NC membrane was marked as T1, the antibody that could detect the Omicron strain was marked as T2, and the control line was still marked as C. According to the results of the two detection lines, the identification effect of Omicron mutant was achieved.

The four mutant virus cultures were diluted to strongly positive (1000 TCID_50_/mL) and weakly positive (250 TCID_50_/mL) concentrations, and the test was repeated twice with the above detection cards to confirm the feasibility of the test strip. Then, the original strain, β, δ mutant strains were mixed with the O strain successively and tested to confirm the detection effect.

### 2.5. Colloidal Gold Strip Cross-Reaction Detection

The detection of original strain, β, δ and Omicron variants of SARS-CoV-2 were completed by China Wuhan Blood Products Co., Ltd.

According to the “Guidelines for the registration and Review of novel coronavirus 2019-nCoV antigen detection reagents” [23], the cross-reaction test was carried out with 48 virus samples, including 40 virulent viruses, such as endemic human coronaviruses HKU1, OC43, NL63, 229E, and 8 common respiratory pathogens. The kit designed in this study was verified by cross-reaction.

### 2.6. Anti-Interference Detection of Colloidal Gold Strip

A total of 25 interfering substances were purchased through the national reference material resource platform, including purified mucin, α-interferon, oseltamivir, peramivir, lopinavir, fluticasone, etc. Inactivated viral cultures of the SARS-CoV-2 original strain were diluted as positive samples with lysates containing a negative nasal swab matrix (interfering substances such as drugs and antibiotics may interfere with the test results). In this study, 25 interfering substances were prepared in different concentrations provided in the Table 1, and then added to the above positive samples. 

Finally, the mixed samples were tested on the dipstick strip, and each sample was tested at least three times in duplicate.

### 2.7. The Minimum Detection Limit and Inclusiveness Detection of Colloidal Gold Dipstick

Negative nasal swab samples were mixed in the lysate as required and used as virus dilution medium. Then, 16 types of culture samples of novel coronavirus mutant strains from 4 different sources and lots were diluted in the diluent medium, respectively.

In the minimum detection limit determination experiment, three cultures were selected, and the virus titers were diluted to the following five concentrations: 500 TCID_50_/mL, 250 TCID_50_/mL, 125 TCID_50_/mL, 62.5 TCID_50_/mL and 31.25 TCID_50_/mL for preliminary detection. Another three cultures were selected, and the virus titers were diluted to the following four concentrations: 250 TCID_50_/mL, 125 TCID_50_/mL, 62.5 TCID_50_/mL and 31.25 TCID_50_/mL, and the confirmatory test was performed again. The above dilutions of each viral culture were repeated 3 to 5 times, and each dilution was repeated no less than 20 times. The detection limit value was obtained by counting the detection results.

The virus titers of the remaining 10 cultures were diluted to 1250 TCID_50_/mL, 125 TCID_50_/mL, 62.5 TCID_50_/mL and 31.25 TCID_50_/mL, and the dilutions of each culture were repeated 3 to 5 times, and each dilution was repeated for at least 20 times. A sample of 120 µL was added to each test card, and the results were observed after 15–20 min; those observed after more than 30 min were invalid.

### 2.8. Stability Test of Colloidal Gold Dipstick

A part of the prepared test strip was placed at 45 °C for accelerated aging test, and random sampling was performed every 7 days till the 56th day. The other part was stored at room temperature and low temperature for real-time stability experiment, and random sampling was performed every 3 months. The samples were tested with appropriate concentrations of positive and negative samples, and each sample was tested in duplicate at least three times.

### 2.9. Study on Positive Judgment Value of Colloidal Gold Dipstick 

Our team carried out positive test in Chengdu Public Health Clinical Medical Center and Guang ‘an People’s Hospital of Sichuan Province simultaneously.

Oral and throat swab and nose swab samples were collected from no less than 90 subjects infected with novel coronavirus (cases within 7 days after positive for novel coronavirus) in Chengdu. Professional researchers collected 1 swab sample and 1 oral and pharyngeal swab sample from the subjects. Nasal swab samples were tested with this strip, and oral and pharyngeal swab samples were tested with the novel coronavirus 2019-nCoV nucleic acid detection kit (fluorescent PCR method) produced by Sun Yat-sen University Daan Gene Co., Ltd. (Guangzhou, China). The consistency of test reagent and nucleic acid reagent was evaluated.

Nasal and oropharyngeal swabs were collected from more than 120 NCoV-negative subjects at Guang’an People’s Hospital in Sichuan Province. Nasal swab samples were tested with this strip, and oral and pharyngeal swab samples were tested with the latest novel coronavirus 2019-nCoV nucleic acid detection kit (fluorescent PCR method) produced by Michael Bio Co., Ltd. (Chengdu, China). to evaluate the consistency of the assessment reagent and nucleic acid reagent detection results.

All the data of the two institutions were collected to complete the study on the positive judgment value of this colloidal gold dipstick.

## 3. Results

### 3.1. Screening Results of Anti-N Protein Monoclonal Antibodies

Monoclonal antibodies specific for SARS-CoV-2 were screened by cell fusion assay and indirect Elisa. The recombinant N protein was used for titer determination in the experiment. When the antibody was diluted to 512,000 times and the S/N value was greater than 2.0, the result was considered as positive. Seven antibodies were finally screened out, and the Elisa results are shown in Table 2.

### 3.2. Screening Results of Colloidal Gold Dipstick

After antibody pair matching, the antigen of different mutant strains was detected, and the test results are shown in Table 3. Antibody pair 4 that could detect the Omicron mutant strain was screened out, and the antibodies used were Ab 1 and Ab 4 in Table 2 above. Several antibody pairs were able to detect the original strain, β and δ mutant strains of SARS-CoV-2, but were negative for Omicron mutant strain. Antibody pair 6 was the most sensitive, and the antibodies used were Ab 3 and Ab 6 in Table 2. These two antibody pairs were selected for subsequent experiments. 

### 3.3. Sensitivity of Colloidal Gold Test Paper for Detecting α, β, δ, and ο Mutants

Sensitivity of colloidal gold test cards to detect SARS-CoV-2 antigen. The original virus cultures were detected by the scientific research and Development Department of China Wuhan Blood Products Co., Ltd. The detection limit was 1:2 × 10^7^. The detection limit of the Omicron mutant virus culture was 1:1.6 × 10^5^ (the concentration of original culture was 6.2 lgTCID_50_/mL).

According to the instructions of the nucleic acid PCR kit, the inactivated SARS-CoV-2 culture was weakly positive when diluted to 1:10^7^, and negative when diluted to 1:10^8^, and the detection limit was 1:10^7^, as shown in Table 4. The comparison of the sensitivity of antigen colloidal gold detection card showed that the sensitivity of colloidal gold detection card could reach the same level as that of nucleic acid detection reagent [22].

### 3.4. Single-Card Double-T Strip Test for Omicron Mutant Strain

According to the results of two different concentration samples, it was determined that the T2 detection line could be completely visible when the test strip was used to detect a single mutant sample, but the T1 detection line could not appear when the test strip was used to detect the Omicron mutant. Based on these results, the Omicron mutant strain was specifically identified. However, for the two mixed mutant strains, the two detection lines could appear normally at different concentrations and did not produce false positive results (Figure 1).

### 3.5. Colloidal Gold Dipstick Cross-Reaction Verification

According to the test results in Table 5, it can be seen that 48 viruses and other pathogens had no cross interference in the detection of the kit. 

### 3.6. Anti-Interference Test Results of Colloidal Gold Dipstick

According to Table 6 of the test results, the positive test results of this kit for 25 interfering substances were normal under the experimental concentration, and there were no false negatives or strong positives, which proved that these substances had no interference effect. 

### 3.7. Verification of the Minimum Detection Limit and Inclusiveness of Colloidal Gold Dipstick

The detection limit of the test strip was determined and verified. The statistical results are shown in Table 7. The detection limit of the test strip for the original strain, β strain, δ strain and o strain was 62.5 TCID_50_/mL. 

The minimum detection limit of the colloidal gold kit for several virus strains was 62.5 TCID_50_/mL, which showed that the kit products had good inclusiveness for samples of different SARS-CoV-2 strains.

### 3.8. Stability Verification of the Test Strip for SARS-CoV-2 Antigen Detection

The detection of SARS-CoV-2 diluted culture showed that with the extension of storage time, the color of the test line and quality control line of each sample of the test strip was basically unchanged. In the accelerated aging test, the strips showed normal color when stored at 45 °C for 21 days. The real-time stability test strip has been stored for more than 1 year, and the color development is normal.

### 3.9. Study on the Positive Value of Colloidal Gold Strip 

According to statistics, 98 subjects participated in the study in Chengdu Public Health Clinical Medical Center, and the results are shown in Table 8. There were no false positives, but there were 10.2% of false negatives. A total of 138 subjects participated in the study in Guang’an People’s Hospital of Sichuan Province. Statistics showed no false positive or false negative results. 

## 4. Discussion

β-propiolactone has a wide modification effect on polar amino acids on the surface of viral proteins, and then affects the antigenicity and immunogenicity of proteins [24]. In this study, β-propiolactone was used to modify the SARS-CoV-2 N protein antigen to artificially improve the diversity of antigen. In the monoclonal antibody screening stage, synthetic peptide chains, N protein antigen, and inactivated virus were used to test the binding ability of antibodies. As expected, in our results, many antibodies with different binding abilities were obtained. 

Among the antibodies obtained, the binding ability of the antibodies to the inactivated virus was high, the antibody titer detected by the coated N protein antigen was the highest, and the antibody titer detected by the synthetic peptide chain was very low. These results indicate that the Mabs screened in this study mainly bind to the spatial epitopes of the N protein antigen rather than to the nonlinear epitopes.

During the dipstick screening phase, we generated antibody pairs of previous variants that were 1600 times more sensitive than Omicron strain. Existing studies have shown that the Omicron mutant strains have four amino acid mutations in the N protein sequence. The results of this study speculated that the mutation caused a change in the spatial structure of the antigen and a great change in the immune response sensitivity. It is concluded that the great difference in immunogenicity between Omicron mutant and previous mutants can be used as the basis for immunological classification. At the same time, this huge difference may also be a reason for the variation of viral pathogenicity and infectivity, which is worth expanding research for.

Taking advantage of the large difference in the binding affinity of the Mabs to different SARS-CoV-2 mutants, we developed a test strip that can identify Omicron mutants. The strip enables testers to quickly identify patients infected with Omicron mutant strain, which is of great guiding significance for the formulation of isolation measures and treatment plans for SARS-CoV-2 epidemic. In addition, the monoclonal antibodies used in the developed kit showed strong recognition of Omicron mutant strain, and the kit had high detection sensitivity to all various common mutants. This kit could achieve rapid detection of asymptomatic carriers with low viral load and overcome the low positive rate of commercial kits for Omicron mutant strains. Based on the double-antibody sandwich method, immunochromatography and colloidal gold surface antibody directed labeling technology, the highly sensitive SARS-CoV-2 N antigen detection kit was developed, which realized the qualitative detection of SARS-CoV-2 N antigen in samples. The developed product solved the defects of the current detection kits that could not detect the genotype of infection and missed the detection of some important mutants. It has the advantages of fast, simple, high sensitivity and strong specificity.

At the same time, we compared the existing kits with those in the Chinese market and determined that the sensitivity of some kits for detecting Omicron mutant strains also decreased, which was similar to the results of some of our antibodies. This indicates that the antibody screened with wild-type N protein generally has a problem of decreased sensitivity to Omicron mutants, and the screened antibodies must be tested with the Omicron strains. These approved commercial kits should be further improved.

## 5. Conclusions

In this study, we used β-propiolactone to treat N protein of Omicron SARS-CoV-2 mutant strains antigen to immunize mice and performed cell fusion and monoclonal antibody screening experiments. A variety of strong positive antibodies was obtained. Based on this, this study designed a single card double marking test form, that is, two T lines were drawn on a test strip, so that it could detect and identify Omicron SARS-CoV-2 mutant strains.

The sensitivity of the SARS-CoV-2 antigen detection kit can reach 62.5 TCID_50_/mL, and it has good inclusiveness for different SARS-CoV-2 strains. The kit had no cross-reaction with common respiratory pathogens, and its sensitivity was still not affected under the action of different concentrations of interferences, indicating that it had good specificity and stability. According to the test results from Wuhan Institute of Blood Products, the sensitivity of the kit was similar to that of the nucleic acid test, indicating that compared with the traditional microbial detection method and nucleic acid detection method, the SARS-CoV-2 antigen detection kit prepared in this study has the advantages of being rapid and efficient, and meets the requirements of clinical application.

## Figures and Tables

**Figure 1 vaccines-11-00303-f001:**
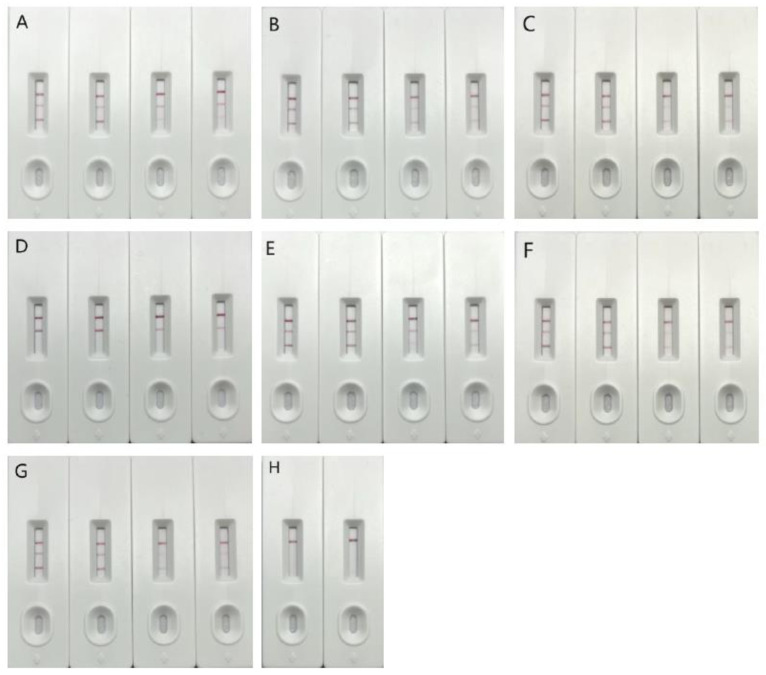
Detection result of single card with double T test line. Note: In the above detection results, (**A**) detection of the original strain, (**B**) detection of the β strain, (**C**) detection of the δ strain, (**D**) detection of the o strain, (**E**) detection of the original strain mixed with o, (**F**) detection of the β strain mixed with o, (**G**) detection of the δ strain mixed with o, (**H**) detection of the negative control. The three detection lines in the figure are, respectively, C, T2 and T1 lines from top to bottom, where C is the quality control line, T line is the detection line, T1 is the undetectable Omicron line, and T2 is the detectable Omicron line.

**Table 1 vaccines-11-00303-t001:** Concentration table of interfering substances.

Interfering Substance	Concentration	Interfering Substance	Concentration	Interfering Substance	Concentration
Mucin	0.1%	Oxymetazoline	200 µg/mL	Levofloxacin	6.27 µg/mL
	0.05%		100 µg/mL		3.14 µg/mL
Alpha interferon	1.57 mg/mL	Histamine hydrochloride	200 µg/mL	Azithromycin	5.23 µg/mL
	3.14 mg/mL		100 µg/mL		2.62 µg/mL
Zanamivir	142 ng/mL	Phenylephrine	20 µg/mL	Ceftriaxone	200 µg/mL
	71 ng/mL		10 µg/mL		100 µg/mL
Ribavirin	4 mg/mL	Beclomethasone Dipropionate	0.3 µg/mL	Meropenem	112 µg/mL
	2 mg/mL		0.15 µg/mL		56 µg/mL
Oseltamivir	1.275 mg/mL	Dexamethasone	0.3 µg/mL	Tobramycin	4 µg/mL
	0.638 mg/mL		0.15 µg/mL		2 µg/mL
Palmer peramivir	23 µg/mL	Flunisolide	0.075 µg/mL	Sodium chloride (with preservatives)	0.85%
	11.5 µg/mL		0.3 µg/mL		0.43%
Lopinavir	13.2 µg/mL	Triamcinolone acetonide	0.5 µg/mL	Mometasone	200 µg/mL
	6.6 µg/mL		0.25 µg/mL		100 µg/mL
Ritonavir	53 µg/mL	Budesonide	0.275 µg/mL	Fluticasone	0.3 ng/mL
	26.5 µg/mL		0.138 µg/mL		0.15 ng/mL
Arbidol	658.5 ng/mL				
	329.25 ng/mL				

(mg = milligram µg = microgramme ng = nanogram mL = milliliter).

**Table 2 vaccines-11-00303-t002:** Elisa results of antibody titer detection after purification.

	Dilution	Ab1	Ab2	Ab3	Ab4	Ab5	Ab6	Ab7
1	1:2000	2.9273	2.8347	2.9312	2.8819	2.8361	2.8177	2.8388
2	1:4000	2.8751	2.8585	2.8830	2.8012	2.8494	2.7350	2.8617
3	1:8000	2.7272	2.8057	2.9817	2.8998	2.8072	2.8722	2.8006
4	1:16,000	2.5464	2.0330	2.7113	2.6177	2.2153	2.5728	2.2679
5	1:32,000	2.2422	1.6490	2.4771	2.4834	1.8094	2.4213	1.8755
6	1:64,000	1.8523	1.2831	1.7938	1.7334	1.2693	1.7220	1.2779
7	1:128,000	1.3884	0.7246	1.2932	1.2618	0.7387	1.0763	0.7530
8	1:256,000	1.0423	0.3942	0.8201	0.8069	0.3968	0.7441	0.4063
9	1:512,000	0.8228	0.1650	0.4624	0.4455	0.1683	0.3612	0.1819
10	1:1,024,000	0.6196	0.0826	0.2424	0.2375	0.0821	0.2176	0.0929
11	Blank	0.0702	0.0589	0.0604	0.0534	0.0611	0.0442	0.0608
12	Blank	0.0684	0.0518	0.0512	0.0510	0.0443	0.0462	0.0425

**Table 3 vaccines-11-00303-t003:** Validation results of antibody pairs against the four strains.

	Dilution Ratio	10^2^				10^3^				10^4^				CON
Pairs of Antibody		α	β	δ	ο	α	β	δ	ο	α	β	δ	ο	
1		+++	+++	+++	+	++	++	++	-	-	-	-	-	-
2		+++	+++	+++	-	++	++	++	-	+	+	-	-	-
3		+++	+++	+++	-	++	++	++	-	+	+	-	-	-
4		+++	+++	+++	+++	++	++	++	++	+	+	+	+	-
5		++	++	++	-	+	+	+	-	+	+	+	-	-
6		+++	+++	+++	-	++	++	++	-	+	+	+	-	-

(Note: + indicates positive results ++/+++ indicates a strong positive result; - indicates negative results).

**Table 4 vaccines-11-00303-t004:** Nucleic acid detection of cultures inactivated with different concentrations of SARS-CoV-2.

Dilution	The CT Value of the TEXAS RED Channel (ORF1ab)	The CT Value of the FAM Channel (N)	Results
1:10	16.98	16.87	positive
1:10^2^	19.47	19.71	positive
1:10^3^	25.43	25.77	positive
1:10^4^	26.31	26.50	positive
1:10^5^	29.85	29.92	positive
1:10^6^	33.16	33.56	positive
1:10^7^	35.11	35.66	positive
1:10^8^	37.56	37.24	negative
1:10^9^	37.78	37.98	negative
1:10^10^	37.59	37.73	negative

**Table 5 vaccines-11-00303-t005:** Test strip cross-reaction validation results.

Sample	Detection Result			Sample	Detection Result		
	1	2	3		1	2	3
Coronavirus HKU1	-	-	-	Adenovirus type 4	-	-	-
Coronavirus OC43	-	-	-	Adenovirus type 5	-	-	-
Coronavirus NL63	-	-	-	Adenovirus type 7	-	-	-
Coronavirus 229E	-	-	-	Adenovirus type 55	-	-	-
SARS	-	-	-	Enterovirus group A	-	-	-
MERS	-	-	-	Enterovirus group B	-	-	-
Novel influenza A (H1N1) virus (2009)	-	-	-	Enterovirus group C	-	-	-
Seasonal H1N1 influenza virus	-	-	-	Enterovirus group D	-	-	-
H3N2	-	-	-	Epstein-barr virus	-	-	-
H5N1	-	-	-	Measles virus	-	-	-
H7N9	-	-	-	Human cytomegalovirus	-	-	-
Influenza B Yamagata	-	-	-	Rotavirus	-	-	-
Influenza B Victoria	-	-	-	Norovirus	-	-	-
Parainfluenza virus type I	-	-	-	Mumps virus	-	-	-
Parainfluenza virus type II	-	-	-	Varicella-zoster virus	-	-	-
Parainfluenza virus type III	-	-	-	Human metapneumovirus	-	-	-
Respiratory syncytial virus type A	-	-	-	Mycoplasma pneumoniae	-	-	-
Respiratory syncytial virus type B	-	-	-	Chlamydia pneumoniae	-	-	-
Human rhinovirus group A	-	-	-	Haemophilus influenzae	-	-	-
Human rhinovirus group B	-	-	-	Staphylococcus aureus	-	-	-
Human rhinovirus group C	-	-	-	Streptococcus pneumoniae	-	-	-
Adenovirus type 1	-	-	-	Klebsiella pneumoniae	-	-	-
Adenovirus type 2	-	-	-	Mycobacterium tuberculosis	-	-	-
Adenovirus type 3	-	-	-	Candida albicans	-	-	-

**Table 6 vaccines-11-00303-t006:** Influence of interfering substances on positive results.

Sample	Concentration	Detection Result			Sample	Concentration	Detection Result		
		1	2	3			1	2	3
Mucin	0.1%	+	+	+	Dexamethasone	0.3 ng/mL	+	+	+
	0.05%	+	+	+		0.15 ng/mL	+	+	+
Alpha interferon	1.57 mg/mL	+	+	+	Flunisolide	0.075 ng/mL	+	+	+
	3.14 mg/mL	+	+	+		0.3 ng/mL	+	+	+
Zanamivir	142 ng/mL	+	+	+	Triamcinolone acetonide	0.5 ng/mL	+	+	+
	71 ng/mL	+	+	+		0.25 ng/mL	+	+	+
Ribavirin	4 mg/mL	+	+	+	Budesonide	0.275 µg/L	+	+	+
	2 mg/mL	+	+	+		0.138 µg/L	+	+	+
Oseltamivir	1.275 mg/mL	+	+	+	Levofloxacin	6.27 µg/mL	+	+	+
	0.638 mg/mL	+	+	+		3.14 µg/mL	+	+	+
Palmer peramivir	23 µg/mL	+	+	+	Azithromycin	5.23 µg/mL	+	+	+
	11.5 µg/mL	+	+	+		2.62 µg/mL	+	+	+
Lopinavir	13.2 µg/mL	+	+	+	Ceftriaxone	200 µg/mL	+	+	+
	6.6 µg/mL	+	+	+		100 µg/mL	+	+	+
Ritonavir	53 µg/mL	+	+	+	Meropenem	112 µg/mL	+	+	+
	26.5 µg/mL	+	+	+		56 µg/mL	+	+	+
Arbidol	658.5 ng/mL	+	+	+	Tobramycin	4 µg/mL	+	+	+
	329.25 ng/mL	+	+	+		2 µg/mL	+	+	+
Oxymetazoline	200 µg/mL	+	+	+	Sodium chloride (with preservatives)	0.85%	+	+	+
	100 µg/mL	+	+	+		0.43%	+	+	+
Histamine hydrochloride	200 µg/mL	+	+	+	Mometasone	200 µg/mL	+	+	+
	100 µg/mL	+	+	+		100 µg/mL	+	+	+
Phenylephrine	20 µg/mL	+	+	+	Fluticasone	0.3 µg/mL	+	+	+
	10 µg/mL	+	+	+		0.15 µg/mL	+	+	+
Beclometasone Dipropionate	0.3 ng/mL	+	+	+					
	0.15 ng/mL	+	+	+					

(mg = milligram µg = microgramme ng = nanogram mL = milliliter).

**Table 7 vaccines-11-00303-t007:** The results of repeated dilution of SARS-CoV-2 culture samples.

Strain Types	Dilution/ TCID50/mL	Positive Numbers	Negative Numbers	Positive Rates	Results
Original	125	400	0	100%	Positive
	62.5	400	0	100%	Positive
	31.25	20	380	5%	Negative
β	125	400	0	100%	Positive
	62.5	400	0	100%	Positive
	31.25	25	375	6.25%	Negative
δ	125	400	0	100%	Positive
	62.5	400	0	100%	Positive
	31.25	12	388	0.25%	Negative
Omicron	125	400	0	100%	Positive
	62.5	400	0	100%	Positive
	31.25	392	8	98%	Positive

(Note: Results with a positive rate greater than 95% were considered positive; mL = milliliter).

**Table 8 vaccines-11-00303-t008:** Comparison table of Chengdu Public Health Clinical Medical Center antigen detection and nucleic acid detection results.

Sample	Antigen Test Result	Nucleic Acid Test Result	ORFlab	N	Sample	Antigen Test Result	Nucleic Acid Test Result	ORFlab	N	Sample	Antigen Test Result	Nucleic Acid Test Result	ORFlab	N
1	+	+	23.25	21.55	34	+	+	16.79	16.29	67	+	+	18.27	17.92
2	+	+	22.23	21.71	35	+	+	16.57	15.91	68	+	+	22.97	21.81
3	+	+	16.08	14.38	36	+	+	19.21	17.49	69	+	+	23.74	22.81
4	+	+	20.2	19.49	37	+	+	17.7	17.5	70	+	+	20.08	19.04
5	+	+	17.65	15.93	38	+	+	18.94	18.13	71	+	+	13.07	11.63
6	+	+	19.58	18.04	39	-	+	21.67	20.24	72	+	+	18.25	17.2
7	-	+	33.61	33.09	40	+	+	21.26	2.06	73	+	+	21.14	20.62
8	+	+	19.49	17.4	41	+	+	16.87	14.77	74	+	+	29.56	29.16
9	+	+	14.11	12.95	42	+	+	15.93	13.8	75	+	+	15.51	14.3
10	+	+	21.96	20.77	43	+	+	14.33	13.12	76	+	+	20.84	19.47
11	+	+	22.41	20.76	44	+	+	25.17	24.98	77	+	+	19.23	18.09
12	+	+	16.55	15.68	45	+	+	14.25	13.36	78	+	+	15.52	14.23
13	-	+	21.03	20.22	46	+	+	21.59	19.99	79	+	+	18.28	16.43
14	+	+	18.22	17.04	47	+	+	20.92	20.4	80	-	+	30.47	29.89
15	+	+	23.12	21.25	48	+	+	17.92	15.85	81	+	+	32.65	32.18
16	+	+	21.27	21.41	49	+	+	24.66	24.11	82	+	+	30.12	28.47
17	+	+	35.21	34.48	50	-	+	26.45	26.36	83	-	+	-	39.3
18	+	+	21.11	20.86	51	+	+	19.14	17.88	84	+	+	30.83	28.87
19	+	+	21.47	21.05	52	+	+	19.19	17.52	85	+	+	33.33	32.79
20	-	+	19.15	17.38	53	+	+	20.94	19.14	86	+	+	36.2	35.17
21	+	+	13.83	11.99	54	+	+	17.35	15.68	87	+	+	32.09	30.84
22	+	+	15.02	12.54	55	+	+	18.9	18.14	88	+	+	31.17	29.61
23	+	+	14.4	12.77	56	+	+	7.67	15.18	89	+	+	37.47	38.09
24	-	+	24.87	23.92	57	+	+	18.2	17.35	90	+	+	23.77	22.38
25	+	+	20.99	20.02	58	+	+	18.5	17.79	91	+	+	29.03	27.83
26	-	+	35.7	35.15	59	+	+	26.35	26.03	92	+	+	30.87	29.09
27	-	+	17.08	15.26	60	+	+	25.42	23.58	93	+	+	27.79	24.8
28	+	+	18.4	17.98	61	+	+	19.16	17.89	94	+	+	30.71	29.54
29	+	+	16.97	16.27	62	+	+	23.89	23.42	95	+	+	16.26	13.19
30	+	+	23.88	23.07	63	+	+	21.7	20.58	96	+	+	32.24	29.38
31	+	+	26.13	26.17	64	+	+	15.85	14.79	97	+	+	32.78	30.95
32	+	+	18.34	17.94	65	+	+	19.16	17.89	98	+	+	33.79	31.44
33	+	+	18.01	17.16	66	+	+	23.74	22.65					

## Data Availability

Data will be made available on request.

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
