# Peer review of "Development of an Antigen Detection Kit Capable of Discriminating the Omicron Mutants of SARS-CoV-2"

_vaccines, 2023, doi:10.3390/vaccines11020303_

Round 1

Reviewer 1 Report

The manuscript entitled “Development of an Antigen Detection Kit Capable of 2 Discriminating the Omicron Mutants of SARS-CoV-2” details the development of rapid antigen detection kit for SARS-Cov 2 that can differentiate between omicron with other strains of the virus. Authors have successfully made monoclonal antibodies against the virus and paired them on colloidal gold-immunochromatography test strips to make the detection strip. They have checked the specificity and selectivity of the test kit against various other pathogens. The prepared kit is highly selective and specific for SARS-Cov-2 viruses. The results of this study can introduce a new antigen detection kit in the market for rapid and efficient SARS-cov2 virus detection.

The abstract is precise and to the point. Overall, the paper is written beautifully and in detail.

The reviewer has a few major and minor comments listed here:

Major Comments:

1.     Is there any study or experiment done to detect the percentage for false negative and false positive result? If yes, then please present the data and if no, please do it.

2.     Authors have said that this kit is as sensitive as PCR test. Images for this data are missing.

3.     Methods- Preparation of monoclonal antibodies against SARS-CoV-2 N protein.

How did the authors get N protein (Source?) information is missing.

Method seems incomplete. Authors have to explain how many different strains N protein was used for raising antibodies. How many mice were used? How many sets of antibodies were made?

Minor Comments:

1.     Line 44; Please add reference.

2.     Line 66; What is the predetermined value. Please write and its reference also.

3.     Line 81; Did the authors culture the virus in lab? If so, please mention the method. If they procured the culture from somewhere else, please mention the reference.

4.     Line 102; Also explain different lines on the test strip (C, T1 and T2) here to avoid confusion further in the manuscript.

5.     Line 178; Which PCR test kit was used? Which antigens do they test?

6.     Line 183; Table 4;

A.   Alongside fluorescence channel also write which antigen they are detecting.

B.    What is S1 culture?

7.     Line 194. Figure1: Why are there four strips in each group?

8.     Line 211; Verification of the minimum detection limit and inclusiveness of colloidal gold dipstick. It would be better to show the data for this experiment in tabular form.

9.     Line 220; ‘quality control line of each sample of t-he test strip’. Needs editing.

10.  Line 237-238; Please rephrase.

11.  Line 239; It would be better of authors can write a little more about the mutations and their location.

Author Response

  1. Reviwer1

    The manuscript entitled “Development of an Antigen Detection Kit Capable of 2 Discriminating the Omicron Mutants of SARS-CoV-2” details the development of rapid antigen detection kit for SARS-Cov 2 that can differentiate between omicron with other strains of the virus. Authors have successfully made monoclonal antibodies against the virus and paired them on colloidal gold-immunochromatography test strips to make the detection strip. They have checked the specificity and selectivity of the test kit against various other pathogens. The prepared kit is highly selective and specific for SARS-Cov-2 viruses. The results of this study can introduce a new antigen detection kit in the market for rapid and efficient SARS-cov2 virus detection.

    The abstract is precise and to the point. Overall, the paper is written beautifully and in detail.

    The reviewer has a few major and minor comments listed here:

    Major Comments:

    1. Is there any study or experiment done to detect the percentage for false negative and false positive result? If yes, then please present the data and if no, please do it.

    Answer: Thanks for the suggestions, our team conducted a positive judgment value study before formal clinical practice, and carried out an experiment in Chengdu Public Health Clinical Medical Center. A total of 98 subjects participated in the study. Statistics showed that there were no false positives, but there were 9.2% false negatives. This result is provided as an attachment and is not presented in the article.

    1. Authors have said that this kit is as sensitive as PCR test. Images for this data are missing.

    Answer: Thanks for the suggestions, the data result is quoted from reference 22 (the laboratory provides the test strips, and Wuhan Institute of Biological Products Co. Ltd is responsible for the test). Due to space, the picture is not directly used.

    1. Methods- Preparation of monoclonal antibodies against SARS-CoV-2 N protein. How did the authors get N protein (Source?) information is missing. Method seems incomplete. Authors have to explain how many different strains N protein was used for raising antibodies. How many mice were used? How many sets of antibodies were made?"

    Answer: Thanks for the suggestions, at the initial stage of screening, antigens produced by Nanjing Jinsilui Biotechnology Co., LTD. (article no. : T20032021) and Wuhanjian Biotechnology Co., LTD. (Article No. : 618-092320-B01) were purchased. We have added the related information in the introduction. (lines66) After experiments, hundreds of mice were used to determine the effectiveness of antigens provided by Pujian Biotechnology Co., LTD. Finally, 7 strains of high-valence antibodies were selected for mutual combination, and 6 groups of effective antibody sets were finally selected. We have added the related information in the introduction(lines175 and 184).

    Minor Comments:

    1. Line 44; Please add reference.

    Answer: Thanks for the suggestions, we have added the related information in the introduction. (lines44) 

    1. Line 66; What is the predetermined value. Please write and its reference also.

    Answer: Thanks for the suggestions, we have added the related information in the introduction. (lines68-69)

    1. Line 81; Did the authors culture the virus in lab? If so, please mention the method. If they procured the culture from somewhere else, please mention the reference.

     Answer: Thanks for the suggestions, the virus was not cultured in this experiment, and was detected by Wuhan Institute of Biological Products Co. Ltd, and we have added the related information in the introduction. (lines106)

    1. Line 102; Also explain different lines on the test strip (C, T1 and T2) here to avoid confusion further in the manuscript.

    Answer: Thanks for the suggestions, we have added the related information in the introduction. (lines111-114)

    1. Line 178; Which PCR test kit was used? Which antigens do they test?

    Answer: Thanks for the suggestions, the actual test is for the antigen of the virus culture, without using a PCR kit.

    1. Line 183; Table 4;
    2. Alongside fluorescence channel also write which antigen they are detecting.

     Answer: Thanks for the suggestions, we have added the related information in the introduction. (lines199)

    1. What is S1 culture?

     Answer: Thanks for the suggestions, it's the original virus culture. To prevent confusion has been removed. We have added the related information in the introduction. (lines198)

    1. Line 194. Figure1: Why are there four strips in each group?

     Answer: Thanks for the suggestions, we have added the related information in the introduction. (lines117-118)

    1. Line 211; Verification of the minimum detection limit and inclusiveness of colloidal gold dipstick. It would be better to show the data for this experiment in tabular form.

     Answer: Thanks for the suggestions, we have added the related information in the introduction. (lines234-235)

    1. Line 220; ‘quality control line of each sample of t-he test strip’. Needs editing.

     Answer: Thanks for the suggestions, we have added the related information in the introduction. (lines238-239)

    1. Line 237-238; Please rephrase.

     Answer: Thanks for the suggestions, we have added the related information in the introduction. (lines246-253)

    1. Line 239; It would be better of authors can write a little more about the mutations and their location.

     Answer: Thanks for your suggestion. Due to limited time, relevant research has not been done in this paper and no modification has been made.

    Reviewer 2

    This is an interesting study in which the Authors describe the development of a rapid antigen test that can detect the SARS-CoV-2 virus, as well as identify whether or not the detected virus belongs to the Omicron complex.

    There does not appear to be cross-reactivity between detection of the SARS-CoV-2 virus, Omicron variant, and other common viruses and bacteria that cause respiratory tract infections.  In addition, the sensitivity of the antigen assay had similar sensitivity to a commercial PCR assay. 

    Major Comments:

    1.The authors discuss that this rapid antigen assay is able to detect the SARS-CoV-2 virus as well as the Omicron variant.  It would be helpful if the Authors discussed which Omicron subvariants were tested.  There have been numerous Omicron subvariants that have circulated across the globe over the past year.  Which Omicron subvariants do the Authors think that this assay can detect?

    Answer: Thanks for your suggestion. This kit has been used commercially in China, and can effectively detect the circulating strains of Omicron such as ba.2, ba.5,ba.7 and xbb that have appeared in the past year.

    2.Although the Omicron variant has been circulating for over a year, as the Authors point out, other variants preceded the Omicron variant.  It would be interesting to hear the Authors perspective on how useful an Omicron-specific rapid antigen test is, given that a new variant may eventually outcompete the Omicron variant.

    Answer: Thanks for your suggestion. In the existing population, coronavirus can be divided into two branches, and the novel coronavirus can also be divided into two categories after the emergence of the Omicron variant. The strain before the Omicron variant is more virulent and less transmissible, while the strain after the Omicron variant is less virulent but more transmissible. Immunological methods can be used for classification and identification, which is helpful for rational drug use. And even hybrids of Delta and Omicron have appeared in other parts of the world, such as XBC, XAY, and XAW. Identifying whether there is a mixed infection is of great help in getting prompt treatment.

    3.Similarly, the prevention and treatment of COVID-19 infection is similar regardless of the variant causing infection (for the most part. The use of monoclonal antibodies does differ depending on the variant). Outside of surveillance purposes, what benefits do the Authors think that an Omicron-specific antigen assay provides?

    Answer: Thanks for your suggestion. As China gradually lifts its policy on COVID-19, antigen testing has become an important means of regular epidemic prevention. As we all know, Omicron not only has S protein variation, but also has structural variation in n protein, so we must pay attention to the effectiveness of the current antigen detection kit to detect mutant strains. Even if COVID-19 mutates, new antibodies can be developed in time with our technology.

    Minor Comments:

    1.References #1 and #6 are duplicates

    Answer: Thanks for the suggestions, we have added the related information in the introduction. (lines321-322)

    2.Reference #22 is incomplete 

    Answer: Thanks for the suggestions, we have added the related information in the introduction. (lines356-357)

    1. All Tables should give the full name of all abbreviations, below the Table. For example, mL = milliliter. 

    Answer: Thanks for the suggestions, we have added the related information in the introduction. (line140)

Reviewer 2 Report

This is an interesting study in which the Authors describe the development of a rapid antigen test that can detect the SARS-CoV-2 virus, as well as identify whether or not the detected virus belongs to the Omicron complex.

There does not appear to be cross-reactivity between detection of the SARS-CoV-2 virus, Omicron variant, and other common viruses and bacteria that cause respiratory tract infections.  In addition, the sensitivity of the antigen assay had similar sensitivity to a commercial PCR assay. 

Major Comments:

1. The authors discuss that this rapid antigen assay is able to detect the SARS-CoV-2 virus as well as the Omicron variant.  It would be helpful if the Authors discussed which Omicron subvariants were tested.  There have been numerous Omicron subvariants that have circulated across the globe over the past year.  Which Omicron subvariants do the Authors think that this assay can detect?

2. Although the Omicron variant has been circulating for over a year, as the Authors point out, other variants preceded the Omicron variant.  It would be interesting to hear the Authors perspective on how useful an Omicron-specific rapid antigen test is, given that a new variant may eventually outcompete the Omicron variant.

3. Similarly, the prevention and treatment of COVID-19 infection is similar regardless of the variant causing infection (for the most part. The use of monoclonal antibodies does differ depending on the variant). Outside of surveillance purposes, what benefits do the Authors think that an Omicron-specific antigen assay provides?

Minor Comments:

1. References #1 and #6 are duplicates

2. Reference #22 is incomplete 

3. All Tables should give the full name of all abbreviations, below the Table. For example, mL = milliliter. 

Author Response

Major Comments:

  1. The authors discuss that this rapid antigen assay is able to detect the SARS-CoV-2 virus as well as the Omicron variant.  It would be helpful if the Authors discussed which Omicron subvariants were tested.  There have been numerous Omicron subvariants that have circulated across the globe over the past year.  Which Omicron subvariants do the Authors think that this assay can detect?

Answer: Thanks for your suggestion. This kit has been used commercially in China, and can effectively detect the circulating strains of Omicron such as ba.2, ba.5,ba.7 and xbb that have appeared in the past year.

  1. Although the Omicron variant has been circulating for over a year, as the Authors point out, other variants preceded the Omicron variant.  It would be interesting to hear the Authors perspective on how useful an Omicron-specific rapid antigen test is, given that a new variant may eventually outcompete the Omicron variant.

Answer: Thanks for your suggestion. In the existing population, coronavirus can be divided into two branches, and the novel coronavirus can also be divided into two categories after the emergence of the Omicron variant. The strain before the Omicron variant is more virulent and less transmissible, while the strain after the Omicron variant is less virulent but more transmissible. Immunological methods can be used for classification and identification, which is helpful for rational drug use. And since variant hybrids of Delta and Omicron have appeared in other parts of the world, identifying mixed infections could be a big help in getting treatment

  1. Similarly, the prevention and treatment of COVID-19 infection is similar regardless of the variant causing infection (for the most part. The use of monoclonal antibodies does differ depending on the variant). Outside of surveillance purposes, what benefits do the Authors think that an Omicron-specific antigen assay provides?

Answer: Thanks for your suggestion. As China gradually lifts its policy on COVID-19, antigen testing has become an important means of regular epidemic prevention. As we all know, Omicron not only has S protein variation, but also has structural variation in n protein, so we must pay attention to the effectiveness of the current antigen detection kit to detect mutant strains. Even if COVID-19 mutates, new antibodies can be developed in time with our technology.

Minor Comments:

  1. References #1 and #6 are duplicates

Answer: Thanks for the suggestions, we have added the related information in the introduction. (lines320-321)

  1. Reference #22 is incomplete 

Answer: Thanks for the suggestions, we have added the related information in the introduction. (lines355-356)

  1. All Tables should give the full name of all abbreviations, below the Table. For example, mL = milliliter. 

Answer: Thanks for the suggestions, we have added the related information in the introduction. (lines139)

Round 2

Reviewer 1 Report

1.Authors have made the changes as suggested but they are not presented in the corrections at correct positions. It is very difficult to see where the authors are pointing and where the correction is done.

2. I think the result for false positive and false negative should be mentioned in the manuscript.

Author Response

Reviewer1:

The manuscript entitled “Development of an Antigen Detection Kit Capable of 2 Discriminating the Omicron Mutants of SARS-CoV-2” details the development of rapid antigen detection kit for SARS-Cov 2 that can differentiate between omicron with other strains of the virus. Authors have successfully made monoclonal antibodies against the virus and paired them on colloidal gold-immunochromatography test strips to make the detection strip. They have checked the specificity and selectivity of the test kit against various other pathogens. The prepared kit is highly selective and specific for SARS-Cov-2 viruses. The results of this study can introduce a new antigen detection kit in the market for rapid and efficient SARS-cov2 virus detection.

The abstract is precise and to the point. Overall, the paper is written beautifully and in detail.

The reviewer has a few major and minor comments listed here:

Major Comments:

  1. Authors have made the changes as suggested but they are not presented in the corrections at correct positions. It is very difficult to see where the authors are pointing and where the correction is done.

Answer: Thanks for the suggestions, due to the adjustment of the article section, the revised opinions cannot correspond to the revised content correctly. The whole article has been readjusted and the reply has been made. All changes have been marked in yellow.

  1. I think the result for false positive and false negative should be mentioned in the manuscript.

Answer: Thanks for the suggestions, we have added the related information in the introduction. (lines168-185 and 261-266)

1.Is there any study or experiment done to detect the percentage for false negative and false positive result? If yes, then please present the data and if no, please do it.

Answer: Thanks for the suggestions, we have added the related information in the introduction. (lines163-180 and 255-264)

2.Authors have said that this kit is as sensitive as PCR test. Images for this data are missing.

Answer: Thanks for the suggestions, the data result is quoted from reference 22 (the laboratory provides the test strips, and Wuhan Institute of Biological Products Co. Ltd is responsible for the test). Due to space, the picture is not directly used.

3.Methods- Preparation of monoclonal antibodies against SARS-CoV-2 N protein. How did the authors get N protein (Source?) information is missing. Method seems incomplete. Authors have to explain how many different strains N protein was used for raising antibodies. How many mice were used? How many sets of antibodies were made?"

Answer: Thanks for the suggestions, at the initial stage of screening, antigens produced by Nanjing Jinsilui Biotechnology Co., LTD. (article no. : T20032021) and Wuhanjian Biotechnology Co., LTD. (Article No. : 618-092320-B01) were purchased. We have added the related information in the introduction. (lines65) After experiments, hundreds of mice were used to determine the effectiveness of antigens provided by Pujian Biotechnology Co., LTD. Finally, 7 strains of high-valence antibodies were selected for mutual combination, and 6 groups of effective antibody sets were finally selected. We have added the related information in the introduction(lines188 and 197).

Minor Comments:

  1. Line 44; Please add reference.

Answer: Thanks for the suggestions, we have added the related information in the introduction. (lines44)  

  1. Line 66; What is the predetermined value. Please write and its reference also. 

Answer: Thanks for the suggestions, we have added the related information in the introduction. (lines67-68)

  1. Line 81; Did the authors culture the virus in lab? If so, please mention the method. If they procured the culture from somewhere else, please mention the reference.

 Answer: Thanks for the suggestions, the virus was not cultured in this experiment, and was detected by Wuhan Institute of Biological Products Co. Ltd, and we have added the related information in the introduction. (line 103)

  1. Line 102; Also explain different lines on the test strip (C, T1 and T2) here to avoid confusion further in the manuscript.      

Answer: Thanks for the suggestions, we have added the related information in the introduction. (lines108-111)

  1. Line 178; Which PCR test kit was used? Which antigens do they test?

Answer: Thanks for the suggestions, the actual test is for the antigen of the virus culture, without using a PCR kit.

  1. Line 183; Table 4;
  2. Alongside fluorescence channel also write which antigen they are detecting.

 Answer: Thanks for the suggestions, we have added the related information in the introduction. (line 211)

  1. What is S1 culture?

 Answer: Thanks for the suggestions, it's the original virus culture. To prevent confusion has been removed. We have added the related information in the introduction. (line 210)

  1. Line 194. Figure1: Why are there four strips in each group?

 Answer: Thanks for the suggestions, we have added the related information in the introduction. (lines114-115)

  1. Line 211; Verification of the minimum detection limit and inclusiveness of colloidal gold dipstick. It would be better to show the data for this experiment in tabular form.

 Answer: Thanks for the suggestions, we have added the related information in the introduction. (lines246-247)

  1. Line 220; ‘quality control line of each sample of t-he test strip’. Needs editing.

 Answer: Thanks for the suggestions, we have added the related information in the introduction. (line 250)

  1. Line 237-238; Please rephrase.

 Answer: Thanks for the suggestions, we have added the related information in the introduction. (lines269-275)

  1. Line 239; It would be better of authors can write a little more about the mutations and their location.

 Answer: Thanks for your suggestion. Due to limited time, relevant research has not been done in this paper and no modification has been made.

Reviewer 2 Report

Thank you for your responses and clarifications to my queries.  

Author Response

Thanks for the comment